# Comparative Evaluation of the Nutritive, Mineral, and Antinutritive Composition of *Musa sinensis* L. (Banana) and *Musa paradisiaca* L. (Plantain) Fruit Compartments

**DOI:** 10.3390/plants8120598

**Published:** 2019-12-12

**Authors:** Barnabas Oluwatomide Oyeyinka, Anthony Jide Afolayan

**Affiliations:** Medicinal Plants and Economic Development (MPED) Research Centre, Botany Department, University of Fort Hare, Alice 5700, South Africa; barnabastom@yahoo.com

**Keywords:** antinutrients, medicinal plants, mineral elements, *Musa sinensis* L., *Musa paradisiaca* L., plant metabolites

## Abstract

Banana and plantain contribute significantly to food security and amelioration of malnutrition, earning their status as staples in several localities of tropical and sub-tropical regions. The distribution of metabolites within the various parts also remains as a key essential to their nutritive and therapeutic potential. This study was aimed at evaluating the nutritional and mineral composition of the flesh, peel, and peel extract components of *Musa sinensis* L. and *Musa paradisiaca* L. fruits as well as their nutritional and therapeutic potentials. Proximate and antinutritional analyses were carried out using standard analytical methods of the Association of Official Analytical Chemists (AOAC), while the mineral constituents were evaluated using inductively coupled plasma optical emission spectroscopy (ICP-OES). Proximate analysis revealed that the flesh and peel of *M. sinensis* L. and *M. paradisiaca* L. contain substantial amounts of moisture, fiber, carbohydrates, and low fat content, while minerals K, Mg, Ca, Na, P, and N were substantially concentrated in the peels and peel extracts in particular. The antinutrients alkaloid, oxalate, saponin, and phytate were detected in safe amounts according to the World Health Organization (WHO). The study points out that the peel and its derivative extract, as well as the flesh of *M. sinensis* L. and *M. paradisiaca* L. are to be put to more relevant human nutritional and therapeutic use.

## 1. Introduction

Fruits are natural sources of useful fiber, minerals, and vitamins. The United Nations Children’s Fund has mirrored the state of affairs of global food security and nutrition, with the sub-regions of Africa chiefly implicated in rising hunger and economic downturn. The dwindling nutritional health in high school children in the past two decades has been highlighted and related to the absence of regular fruit intake in diets [1,2]. The WHO recognizes and prioritizes the frontiers of food security, nutrition improvement, and hunger eradication including agricultural sustainability [3]. Improved global fruit supply, similar to vegetables and pulses, remains key to the reduction in under-nutrition and obesity conditions [4,5].

The prevalence of global mineral malnutrition has been worryingly highlighted, with specific examples including zinc, iron, calcium, copper, and magnesium [6,7]. Several key minerals to human nutrition are sourced chiefly from plants [7,8].

Primary metabolites from plants like fatty acids, carbohydrates, dietary fiber, and amino acids are central in the developmental and physiological processes in plants, while secondary metabolites (e.g., phenolics, carotenoids, sterols) are involved in plant protection and pollinator attraction [9,10].

The banana (*Musa sinensis* L.) and plantain (*Musa paradisiaca* L.) belong to the Musaceae family and have been available for human use for ages [11,12]. They have marked morphological similarities and are of economic and nutritional value. Banana and plantain are majorly tropical plants, but do not grow in temperate regions. They grow predominantly in the continents of Asia, South America, and the tropical African sub-continent [13]. *M. sinensis* and *M. paradisiaca* fruits are essentially good potassium-based dietary items and serve as staples in many global cultures. Potassium is useful in ensuring proper muscular functioning [14]. *M. sinensis* and *M. paradisiaca* fruits are utilized in beverage production, especially among subsistence farmers [15], while other parts, including the roots and flowers are medicinally useful.

The nutritional and therapeutic values of *M. sinensis* and *M. paradisiaca* fruit flesh are undisputedly clear. However, this study was undertaken to evaluate and establish the nutritive, therapeutic, and pharmacological potentials of *M. sinensis* and *M. paradisiaca* flesh, peels, and their derivative dietary product, to improve the dietary acceptability and reduce biological wastage respectively of their peels (exocarp).

## 2. Results

### 2.1. Proximal Content

The proximate content of *M. sinensis* (banana) and *M. paradisiaca* (plantain) representing the nutritive components of their flesh, peel, and boiled peel extracts is depicted in Table 1.

The moisture content ranged from 20.81 ± 0.72% in banana flesh to 20.97 ± 0.19% in banana peel and from 20.92 ± 0.02% in plantain flesh to 21.81 ± 0.04% in plantain peel. The ash content was highest in the boiled peel extracts of banana (6.56 ± 0.00%) and plantain (3.39 ± 0.00%) respectively, while it was lowest in the flesh of banana (1.01 ± 0.00%) and plantain (0.78 ± 0.00%) accordingly. The fat content ranged from 0.15 ± 0.00% in banana flesh to 1.24 ± 0.00% in peel and from 0.24 ± 0.00% in plantain flesh to 1.10 ± 0.00% in peel. Crude fiber content was 0.73 ± 0.00% and 4.17 ± 0.00% in banana flesh and peel respectively, while it was 2.45 ± 0.00% in the boiled peel extract compared to 10.24 ± 0.00% in plantain flesh. Crude protein content was 1.71 ± 0.00% in banana flesh, 2.48 ± 0.00% in banana peel, 1.22 ± 0.00% in plantain flesh, and 2.23 ± 0.00% in plantain peel, but 2.15 ± 0.00% and 1.82 ± 0.00% in the peel extracts of banana and plantain, respectively.

Total carbohydrate content ranged from 67.29 ± 0.19% in banana peel to 88.47 ± 0.00% in banana peel extract, while for plantain, it ranged from 66.60 ± 0.02% in the flesh to 91.57 ± 0.00% in its peel extract. The estimated total energy value for banana flesh and peel was 310.55 ± 0.01% and 290.24 ± 0.01% respectively, while it was 273.44 ± 0.01% and 293.10 ± 0.07% in plantain flesh and peel, respectively.

### 2.2. Mineral Composition

Table 2 shows the comparative mineral content of *M. sinensis* and *M. paradisiaca* fruit flesh, peel, and peel extract.

Calcium content was highest in the peels of both banana (40.92 ± 2.51 mg/100 g) and plantain (45.64 ± 14.89 mg/100 g), while magnesium was highest in banana flesh (29.39 ± 0.95 mg/100 g) and plantain peel extract (26.30 ± 2.51 mg/100 g). Potassium content was highest in peel extracts of banana (2244.70 ± 3.42 mg/100 g) and plantain (1312.63 ± 4.13 mg/100 g), respectively. Sodium content was highest in the peel extract of banana (11.60 ± 0.95 mg/100 g) and the peel of plantain (7.73 ± 0.95 mg/100 g), while phosphorus content was highest in the peel (27.84 ± 1.64 mg/100 g) and peel extract (46.41 ± 28.42 mg/100 g) of banana and plantain, respectively. Similarly, zinc content was highest in the peel (0.41 ± 0.01 mg/100 g) and peel extract (0.78 ± 0.01 mg/100 g) of banana and plantain, respectively. The peels of banana (0.52 ± 0.02 mg/100 g) and plantain (0.20 ± 0.01 mg/100 g) contained the highest manganese content. Copper content was highest in the peel extracts (0.15 ± 0.01 mg/100 g; 0.19 ± 0.01 mg/100 g) of banana and plantain, respectively. Iron content was generally low, with the highest levels in banana peel (0.07 ± 0.00 mg/100 g), while it was absent across plantain samples. Nitrogen content was returned in appreciably and was the highest in the peels of banana (397.15 ± 0.54 mg/100 g) and plantain (357.42 ± 0.40 mg/100 g), respectively.

Generally, mineral content for manganese, copper, zinc, and iron was low in banana and plantain (flesh, peel, and peel extract samples), while calcium, magnesium, potassium, sodium, and phosphorus contents were substantial.

Furthermore, it is observable that the peel samples particularly, along with the boiled peel extracts of banana and plantain are chief stores of a wide spectrum of calcium, potassium, sodium, phosphorus, and zinc.

### 2.3. Antinutritional Composition

From the antinutritional results shown in Table 3, the boiled peel extract of *M. sinensis* contained the highest alkaloid (1.76 ± 1.92%), oxalate (40.2 ± 5.4%), and saponin (8.12 ± 2.46%) contents, while the phytate content was highest in banana peel (2.78 ± 0.33%). On the other hand, *M. paradisiaca* flesh contained the highest alkaloid (0.93 ± 0.00%) and phytate (2.36 ± 0.02%) contents, while the *M. paradisiaca* peel and boiled peel extract had the highest oxalate (22.2 ± 0.53%) and saponin (2.86 ± 1.23%) contents, respectively. This is also indicative of the highest saponin levels in the boiled peel extracts of *M. sinensis* (8.12 ± 2.46%) and *M. paradisiaca* (2.86 ± 1.23%).

## 3. Discussion

Plant metabolites exist in diverse numbers and contribute to the physiological dynamics of growth and development in plants [16], invariably making them useful, natural, and nutritional sources.

Moisture content is an index for the shelf-life span of fruits, with low moisture levels tending towards a longer shelf-life [17,18]. High moisture levels connote nutritive richness, in fruits and vegetables particularly, but hinder storage duration. The moisture content in flesh and peel of *M. sinensis* and *M. paradisiaca* is relatively high and indicative of their short shelf-life and highly perishable nature [19]. High moisture levels were similarly reported in banana.

The ash content is essentially significant in food because the inorganic bulk is linked to mineral element composition and also aids in microbial growth retardation. Ash content was relatively substantial in the peel and peel extract of *M. sinensis* and *M. paradisiaca*. Ash content in the flesh of *M. paradisiaca* was low, a similar report was made by [20] on ash content levels in the plantain cultivars Atagafong and Nibrator, just as in the plantain cultivar Okpoisan reported by [21]. *M. paradisiaca* peel contained relatively low ash content, just as [21] observed in the peel of plantain cultivar Opuasinberiba. *M. sinensis* flesh had low ash content level which is also in coherence with the low levels reported in the Saba banana cultivar [19].

Dietary fiber is an essential component of diets which aids the digestive system and facilitation of bowel movement, including weight management, reduced cardiac risk conditions, and gastro-intestinal health. This thus places *M. sinensis* and *M. paradisiaca* peels as handy roughage sources. More so, the extracts from their peels offer similar therapeutic potentials, which align with scientific reports on the biological activity of ethyl acetate, n-hexane, and ethanolic extracts of banana peels of three varieties of banana from the *Musa* genus [22].

Dietary fat at moderate levels can be useful for healthy nutrition, but in excess, dietary fat consumption is detrimental to health and [23] has attempted to utilize fruit and vegetable intake in order to modify fat levels in the body. The generally low fat levels in *M. sinensis* and *M. paradisiaca* puts the two fruits and their compartments entirely in good stead for ameliorating hyperlipidemic conditions through weight management and reduction of cardiovascular disease risk. *M. sinensis* and *M. paradisiaca* contained low fat contents respectively, which are in alignment with the reported low fat levels in *Musa* ABB (with one chromosomal set), AAB (with two chromosomal sets) and AAA (with three chromosomal sets) varietal groups [24].

Dietary proteins are key elements in body-building processes and tissue repair. There have been reports of emerging evidence of the functional role of dietary proteins in reducing type II diabetes and glucoregulatory mechanism [25]. The veracity of dietary protein has been further emphasized through investigative reports on their capacity to stem age-related diseases [26]. The protein content in *M. sinensis* and *M. paradisiaca,* especially the peels, could make them useful protein supplements in diets, similar to the plantain chutney dish staple in India. The flesh of *M. sinensis* and *M. paradisiaca* both contain relatively low protein content, just as [22] reported in the flesh (pulp) of immature Saba banana and plantain.

Carbohydrates are major sources of energy in the human body [4], and they also are a key ingredient for the brain, an organ that thrives on glucose. The carbohydrate content was high across *M. sinensis* and *M. paradisiaca* fruits. This implies that the fruits could serve as useful energy sources in diets. A study by Adeniji et al. [27] reported similarly high carbohydrate content in five hybrids of the *Musa* cultivar. A study by Khawas and Deka [28] also reported substantially high levels of carbohydrates in the peel of culinary banana (*Musa* ABB with one chromosomal set).

The energy value of each of the compartments of *M. sinensis* and *M. paradisiaca* fruits was quite low, but more substantial energy values can be derived from the combination of the compartments. The relatively low carbohydrate, moderate protein, and low fat levels were figured in the derivative energy values in this study. Energy value for *M. sinensis* L. flesh sample was in correlation with earlier reports of four Indonesian banana varieties which include Berlin (AA two sets of chromosomes), Ambon Hijau (AAA with three chromosomal sets), Raja Bandung (AAB with two chromosomal sets), and Kepok (AAB with two chromosomal sets) [29].

Mineral elements are essential components of nutrition and the key minerals are basically supplied in balanced diets. Their functional roles involve structural, physiological, and metabolic processes in the body.

Calcium is important for optimal bone growth and development and for the heart, muscular system, and nervous system to function properly. It has a recommended daily allowance (RDA) of 1000 mg for adults [30]. Calcium levels are moderate in the flesh of *M. sinensis* and *M. paradisiaca*, making them useful calcium sources. Reports have been published on similar calcium levels in the Williams cultivar, a similar *Musa* species [31].

Magnesium content was moderately quantifiable across the compartments of *M. sinensis* and *M. paradisiaca* fruits. It has a recommended daily allowance (RDA) of 450 mg [32] and is important for cardiac functioning and stemming the early phase of diabetes [33], nerve impulse transmission, detoxification, and bone and teeth structural strength [34]. Magnesium levels in *M. paradisiaca* peel were moderate in quantity, just as [35] observed in ripened peel of plantain.

Potassium content was high in *M. sinensis* and *M. paradisiaca*, particularly in the peel extracts. The recommended daily allowance (RDA) of potassium for adults is 4700 mg [36]. In this regard, *M. sinensis* and *M. paradisiaca* have the capacity to contribute a large chunk, nearly 50% of the potassium RDA. Potassium has functional roles in hypertension management, cardiac efficiency, and physiological processes [9], including water balance regulation [37]. Potassium levels in the flesh and peel of *M. paradisiaca* were comparably richer than the pulp of *Chrysophyllum albidum* fruit [38].

The recommended daily allowance (RDA) for sodium is 1500 mg [39]. It is an important electrolyte that helps in the maintenance of intracellular and extracellular water balance, including osmotic pressure balance regulation. Sodium content was relatively low in *M. sinensis* and *M. paradisiaca*. Reports have it that a sodium–potassium ion ratio below one is useful in lowering high blood pressure. *M. sinensis* and *M. paradisiaca* therefore have the potential to be useful in this regard, serving as dietary supplementation particularly for hypertensive individuals. The high to low content of potassium to sodium is responsible for the low Na+/K+ ratio. This ratio in food is significant particularly with a ratio of less than one, which is reported to be useful in hypertension risk reduction and control, including blood pressure and related conditions [40,41,42]. Sodium content in the flesh of *M. sinensis* and *M. paradisiaca* are comparable to reported sodium content in two samples of the dwarf Brazilian banana cultivar from different locations [31].

Phosphorus improves calcium absorption and strengthens bones and teeth particularly in minors and expectant mothers. It has a recommended daily allowance (RDA) of 200 to 1000 mg [43]. Phosphorus content in *M. sinensis* and *M. paradisiaca* was relatively low and below the recommended daily allowance, but it can contribute up to one-tenth of this level.

Zinc is an important trace element which functions in the body for cellular processes involving cerebral development, behavioral mechanisms, and body repair, especially regarding wound healing [44]. It has a recommended daily allowance (RDA) of 4 to 14 mg (National Health and Medical Research Council, Australia). Zinc content in *M. sinensis* and *M. paradisiaca* was quite low and could suggest supplementation with zinc-rich dietary options. Similarly, corresponding zinc concentrations were reported for the Williams and dwarf Brazilian banana cultivars [31]. Considering the RDA of zinc, banana and plantain could contribute to part of this demand.

Nitrogen is a vital nutrient for biotic organisms, a constituent of biological make up including nucleic acids and amino acids and as such is linked to protein formation. It is a key messenger in muscular relaxation. A number of nitrogenous compounds are also involved in hormonal, neurotransmitter, immune-competence, and peroxidative defensive functions [45]. Nitrogen content was high across *M. sinensis* and *M. paradisiaca*, suggesting that these fruits can be useful in neurological and antioxidative mechanisms of the body.

Iron is an important component of hemoglobin; its deficiency leads to anaemia. Increased catecholamine levels in children leading to abnormal behavior have been linked to iron deficiency and folic acid deficiency [46]. The body requires iron for oxygen transport protein synthesis, in particular hemoglobin and myoglobin, and for forming heme enzymes including other iron-containing enzymes [47]. It is involved in the transport of oxygen from the lungs to the tissues [8]. Iron has a recommended daily allowance (RDA) of 18 mg [48]. Iron content was quite low in *M. sinensis* and *M. paradisiaca* and below the RDA threshold. The iron content in *M. sinensis* in this study is slightly comparable to the iron levels in the ripe (Rr-Robusta) and unripe (Ur-Robusta) banana cultivars [49].

Manganese is a micronutrient that functions as an enzymatic catalyst and co-factor in the synthesis of fatty acids and glycoproteins [50,51]. It also helps in the growth and development of the skeletal structure and prothrombin formation along with vitamin K. It also has a recommended daily allowance (RDA) of 2.30 mg [52]. Manganese content was low in *M. sinensis* and *M. paradisiaca*. Manganese content in *M. sinensis* peel was slightly comparable to the observation of Hassan et al. [53].

Manganese content in *M. paradisiaca* flesh is also comparable to the levels in green mature plantain and ripe plantain [19].

Copper is key in red blood cell regulation and erythrocytic processes. A high intake of copper in the body can cause symptoms such as bloody urine, hepatic organ damage (liver), and epigastric uneasiness. Copper has a recommended daily allowance (RDA) of 1.1 mg for adults (NHMRC), while it was very low in *M. sinensis* and *M. paradisiaca* and also below the RDA threshold, with only about 10% contribution. The copper level in *M. sinensis* flesh in this study is similar to the reported levels in the dwarf Brazilian cultivar [31].

Saponin content was relatively low in *M. sinensis* and *M. paradisiaca*. The safe level limit below 10% [54] was detected in the fruit compartments. This makes their saponin levels non-hazardous. Furthermore, there are reports that suggest saponin to have the capacity to reduce cholesterol via complex formation and act in bile acid release [52]. Saponin also functions in reducing serum blood glucose and ameliorating diabetes mellitus [55]. *M. paradisiaca* flesh in particular contained even safer saponin levels as compared to *Kedrostis africana* tuber as reported by Unuofin et al. [56].

Dietary phytate lowers blood glucose and lipids and is involved in preventing calcium crystallization in the renal organ (kidney) [57]. Phytate content was uniformly low, with long-term levels between 1% to 6% having the potential to interfere with mineral bioavailability. *M. sinensis* flesh had a safe phytate level which was comparable to phytic acid content in *Kedrostis africana* tuber [56].

Oxalate content was relatively high in *M. sinensis* and *M. paradisiaca*. Its effects can be neutralized by heating [58]. Another approach is the *Oxalobacter formigenes* activity which depletes oxalic acid, by inducing oxalate excretion in the colonic zone of the intestine [59]. Nguyen and Savage [60] reported similar oxalate levels to *M. sinensis* and *M. paradisiaca* in the fruits of *Rubus* sp. (Raspberry) and *Phyllanthus emblica*.

Alkaloids are active plant metabolites that are biologically functional in cell activity and active promoters of hemoglobin formation in leukemic cells [61,62]. Its content was low in *M. sinensis* and *M. paradisiaca*, thus suggesting the safety of the fruits. The average alkaloid content in *M. sinensis* is comparable to the alkaloid levels in the nutritionally viable *Vernonia mespilifolia* Less. [63].

## 4. Materials and Methods

### 4.1. Sample Procurement

Fruits of banana (*M. sinensis*) and especially plantain (*M. paradisiaca*) of Indian origin used in this study were obtained from supermarkets in Alice and East London, both located in Amathole District Municipality of the Eastern Cape Province, South Africa. These areas lie at 32°43′28.66″ and 26°34′5.88″ geographical latitude and longitude.

### 4.2. Sample Preparation

The fruits were rinsed properly with distilled water. The soft flesh and peels were sliced and oven-dried (LABOTEC, Durban, South Africa) at 40 °C for 72 h, in order to avoid the loss of volatile secondary metabolites due to high temperature and obtain proper drying of plant samples. Afterwards, the dried samples were subjected to pulverization in a food blender (Hamilton Beach HBF 500 Series, Virginia, USA). Another set of *M. sinensis* and *M. paradisiaca* peels (1000 g/L) were boiled in a water bath (BUCHI B-480) at 80 °C for 20 min, a modification of [64]. The boiled extracts obtained were freeze-dried for 48 h using a freeze dryer (Ceramic Filter Core Drier CD 052).

### 4.3. Proximal Composition

#### 4.3.1. Moisture Content Determination

Moisture content was measured as described by [64]. A dried, empty weighing vessel was weighed (W1) in an oven (LABOTEC-South Africa). Afterwards, the sample was placed in the vessel and weighed (W2) and dried in an oven at 40 °C for 72 h to constant weight. Upon cooling, the samples were reweighed (W3). The percentage of moisture content was derived as:(1)$$\%\;\mathrm{Moisture}\;\mathrm{content}=\frac{W2-W3}{W2-W1}\times 100$$
where, W1 is the weight of the empty vessel, W2 is the weight of the vessel + sample before drying, W3 is the weight of vessel + dried sample.

#### 4.3.2. Ash Content

This was determined using the dry ashing technique [65]. The crucible was dried at 105 °C for 1 h and was weighed after cooling (W1). Then, 2 g of samples were placed in the weighed crucibles and weighed again (W2). The crucible, with the samples was thereafter ashed at 250 °C for 1 h (Furnace E-Range, E300-P4, MET-U-ED South Africa) and cooled, then weighed again (W3).

Ash content was calculated as:(2)$$\%\;\mathrm{Ash}\;\mathrm{content}=\frac{W2-W3}{W2-W1}\times 100$$
where W1 is the weight of the dried crucible, W2 is the weight of crucible + sample, W3 is the weight of crucible + ashed sample.

#### 4.3.3. Crude Protein

Crude protein determination was extrapolated from the total nitrogen content in the sample, as described in the micro Kjeldahl technique [66]. Then, 2 g of sample was digested by boiling 20 mL of conc. H_2_SO_4_ and a digestion tablet in a Kjeldahl flask, until the mixture became clear. The digest was then filtered into a 250 mL volumetric flask with distilled water and was made up to mark. This set up was prepared for the distillation process and ammonia was steam-distilled from the digest to which 50 mL of 45% NaOH solution was added. Then, 150 mL of distillate was collected into a conical flask which contained 100 mL 0.1 N hydrochloric acid. Afterwards, ammonia and hydrochloric acid reacted in the receiving flask. An estimation of nitrogen was done by back-titrating against 2 M sodium hydroxide (NaOH) with methyl orange as an indicator.

Nitrogen content was calculated:(3)$$\frac{\left[\left(\mathrm{mL}\;\mathrm{standard}\;\mathrm{acid}\times N\;\mathrm{of}\;\mathrm{acid}\right)-\left(\mathrm{mL}\;\mathrm{blank}\times N\;\mathrm{of}\;\mathrm{base}\right)\right]-\left(\mathrm{mL}\;\mathrm{std}\;\mathrm{base}\times N\;\mathrm{of}\;\mathrm{base}\right)\times 1.4007}{\mathrm{Weight}\;\mathrm{of}\;\mathrm{sample}\;\left(g\right)}$$
where, N is the normality, 1.4007 is the single factor of nitrogen molecular weight, crude protein = nitrogen content (sample) × 6.25

#### 4.3.4. Crude Fiber 

Dietary fiber content was determined with the modified acid-base digestion approach [67]. Digestion by boiling was carried out on 5 g of sample with 100 mL of 0.25 M sulfuric acid (H_2_SO_4_) solution under reflux for 30 min and then filtered immediately. Several rinses (four) were done on the insoluble matter with boiling water in order to evacuate the remnant acid content. This procedure was done again on the residue with 100 mL of 0.31 M sodium hydroxide (NaOH) solution, after which the final residue was cleansed of base by washing with distilled water. The residue was oven-dried at 100 °C, cooled in a desiccator and then weighed (C1). This weighed sample was incinerated at 550 °C for 2 h and left to cool in a desiccator and weighed (C2).

Crude fiber content was quantified thus:(4)$$\%\;\mathrm{Crude}\;\mathrm{fiber}=\frac{C2-C1}{\mathrm{Weight}\;\mathrm{of}\;\mathrm{Sample}}\times 100$$

#### 4.3.5. Crude Lipid

This was determined with the Soxhlet extraction method described by the Association of Official Analytical Chemists (AOAC) [66,68]. Lipid content was extracted from 5 g of sample with 100 mL petroleum ether. The mixture was then decanted, and the residual lipid component collected in a weighed clean 500 mL glassware (round bottom flask) (W1). Further sample lipid extraction was exhaustively carried out with 100 mL petroleum ether for 24 h. This was filtered and decanted into the round bottom flask. This lipid content was concentrated to a dry state in a steam bath and oven-dried at about 50 °C. The beaker was then weighed again (W2). Lipid was quantitatively determined as follows:(5)$$\%\;\mathrm{Crude}\;\mathrm{lipid}=\frac{W2-W1}{\mathrm{Weight}\;\mathrm{of}\;\mathrm{original}\;\mathrm{sample}}\times 100$$

#### 4.3.6. Total Carbohydrate Content

Carbohydrate component was obtained by deducting total protein, crude fiber, lipid, and ash content from the total dry matter in the form:
% Total carbohydrate = 100 − (% moisture content + crude fiber + total ash + crude lipid + crude protein).(6)

#### 4.3.7. Energy Value Evaluation

The total energy value of the samples was calculated using the Atwater factors: 4 kcal, 9 kcal, and 4 kcal to determine the caloric value. The summation of the multiplied crude protein, lipid, and carbohydrate values, respectively is shown below:Energy value (kcal/100 g) = (crude protein × 4) + (total carbohydrate × 4) + (crude lipid × 9).(7)

#### 4.3.8. Evaluation of Mineral Element

Analysis of mineral element was carried out to quantitatively evaluate mineral distribution in the fruit segments. The elements (calcium, copper, iron, magnesium, phosphorus, manganese, potassium, zinc, sodium, and nitrogen) were analyzed using the inductively coupled plasma optical emission spectrometer (ICP-OES; Varian 710-ES Series, SMM, Capetown, South Africa).

### 4.4. Determination of Antinutritional Composition

#### 4.4.1. Alkaloid Content

This was determined using the method described by [69]. First, 5 g of pulverized sample was macerated in 200 mL of 10% acetic acid in ethanol. The mixture was covered and left to stand for 4 h. Filtration was done and the filtrate concentrated in a water bath (BUCHI B-480) to one-quarter of the initial volume. Concentrated ammonium hydroxide (NH_4_OH) was then added to the concentrated filtrate in drop-wise flow, until a cloudy fume precipitate was complete. The solution, upon settling, was washed with dilute ammonium hydroxide (NH_4_OH) and filtered with filter paper. The residue was dried and weighed. Alkaloid content was computed as:(8)$$\%\;\mathrm{Alkaloid}=\frac{\mathrm{Weight}\;\mathrm{of}\;\mathrm{precipitate}}{\mathrm{Weight}\;\mathrm{of}\;\mathrm{original}\;\mathrm{sample}}\times 100$$

#### 4.4.2. Oxalate Content

This was determined using the method described by [70]. First, 1 g of pulverized sample was macerated with 75 mL of 3M H_2_SO_4_ (sulfuric acid) in a conical flask. The mixture was stirred and filtered. Then, 25 mL of the filtrate was collected and heated at maintained temperature of 70 °C. The hot aliquot was titrated steadily against 0.05 M KMnO_4_ (potassium permanganate), until an extremely faint, pale pink end point color persistence was observed for 15 s.

Oxalate content was computed as: 1 mL of 0.05 M KMnO_4_ as equivalent to 2.2 mg oxalate or (2.2 mg × titer value).

#### 4.4.3. Phytate (Phytic Acid) Content

Phytic acid content was analyzed using the method as described in [71]. First, 2 g of pulverized sample was weighed into a 250 mL conical flask. This sample was then macerated with 2% hydrochloric acid (HCl) for 3 h. The mixture was filtered using the vacuum pump and filtrating unit with filter paper (Whatman No. 1). Then, 25 mL filtrate aliquot was placed in another 250 mL conical flask with 5 mL of the indicator 0.03% ammonium thiocyanate solution and 53.5 mL distilled water also added. The mixture was subsequently titrated against standard iron (III) chloride solution (0.00195 g iron per mL) until a brownish yellow endpoint color persisted for about five minutes.
% Phytic acid = Titer value × 0.00195 × 1.19 × 100.(9)

#### 4.4.4. Saponin Content

This analysis was carried out following the method described by [72]. First, 1 g of pulverized sample was added to 40 mL of 20% ethanol, homogenized and left for 4 h at 55 °C. This mixture was filtered with vacuum pump coupled with the filtrating unit. Residue was collected and re-extracted with 20 mL of 20% ethanol. Filtrates were concentrated to 40 mL in a water bath (BUCHI B-480) at 90 °C, after which the concentrate was transferred into a separating funnel. Then, 20 mL of diethyl ether was added and vigorously mixed. Afterwards, the lower fraction was collected, with the upper ether layer discarded. Butan-1-ol was added to the re-introduced lower fraction and mixed vigorously again. Then, 5 mL of 5% aqueous sodium chloride (NaCl) was added. The upper fraction (butan-1-ol) was then collected and evaporated in the oven to constant weight.
(10)$$\%\;\mathrm{Saponin}\;=\frac{\mathrm{Weight}\;\mathrm{of}\;\mathrm{fraction}}{\mathrm{Weight}\;\mathrm{of}\;\mathrm{sample}}\times 100$$

### 4.5. Data Analysis

All data were expressed as mean ± standard deviation and subjected to one-way analysis of variance (ANOVA) using the MINITAB 19 statistical package. Statistical differences were measured at *p* < 0.05. Upon analysis, letter variations along the flesh, peel, and peel extract parameters, for each of the *M. sinensis* and *M. paradisiaca* fruits depict significant differences at *p* < 0.05.

## 5. Conclusions

The study revealed that *M. sinensis* and *M. paradisiaca* contained substantial to high levels of potassium, calcium, magnesium, sodium, phosphorus, nitrogen, moisture, and fiber content accordingly. The anti-nutrient contents of saponin, phytate, and alkaloid were within safe limits, with oxalate levels being safe for nutrition through preparatory methods of heat application which include boiling, cooking or steaming.

More notably, the peels and derived extracts of *M. sinensis* and *M. paradisiaca* are especially rich in several macronutrients and should therefore be considered as key human dietary sources of these nutrients, which improve dietary acceptability and eliminate their biological waste.

## Figures and Tables

**Table 1 plants-08-00598-t001:** Proximate composition (%) of banana (*M. sinensis*) and plantain (*M. paradisiaca*) fruit compartments.

	Sample	Moisture	Lipid (Fat)	Ash	Fiber	Protein	Carbohydrate	Energy Value
Banana	Flesh	20.81 ± 0.72 ^a^	0.15 ± 0.00 ^f^	1.01 ± 0.00 ^e^	0.73 ± 0.00 ^f^	1.71 ± 0.00 ^e^	75.59 ± 0.72 ^c^	310.55 ± 0.01 ^c^
	Peel	20.87 ± 0.19 ^ab^	1.24 ± 0.00 ^a^	3.95 ± 0.00 ^b^	4.17 ± 0.00 ^b^	2.48 ± 0.00 ^a^	67.29 ± 0.19 ^e^	290.24 ± 0.01 ^e^
	Peel extract	-	0.75 ± 0.00 ^d^	6.56 ± 0.00 ^a^	2.07 ± 0.00 ^e^	2.15 ± 0.00 ^c^	88.47 ± 0.00 ^b^	369.23 ± 0.01 ^b^
Plantain	Flesh	20.92 ± 0.02 ^ab^	0.24 ± 0.00 ^e^	0.78 ± 0.00 ^f^	10.24 ± 0.00 ^a^	1.22 ± 0.00 ^f^	66.60 ± 0.02 ^f^	273.44 ± 0.01 ^f^
	Peel	21.81 ± 0.04 ^a^	1.10 ± 0.00 ^b^	2.23 ± 0.00 ^d^	4.06 ± 0.00 ^c^	2.23 ± 0.00 ^b^	68.57 ± 0.04 ^d^	293.10 ± 0.07 ^d^
	Peel extract	-	0.77 ± 0.00 ^c^	3.39 ± 0.00 ^c^	2.45 ± 0.00 ^d^	1.82 ± 0.00 ^d^	91.57 ± 0.00 ^a^	381.03 ± 0.01 ^a^

Values indicated are mean ± standard deviation, while letter variations along a column depict significant differences at *p* < 0.05 among samples of *M. sinensis* and *M. paradisiaca* fruits.

**Table 2 plants-08-00598-t002:** Compartmental mineral composition (mg/100 g) in banana (*M. sinensis*) and plantain (*M. paradisiaca*).

	Banana Flesh	Banana Peel	Banana Peel Extract	Plantain Flesh	Plantain Peel	Plantain Peel Extract
Calcium	4.64 ± 1.64 ^d^	40.99 ± 2.51 ^ab^	26.3 ± 2.51 ^bc^	6.96 ± 1.64 ^d^	45.64 ± 14.89 ^a^	15.47 ± 1.89 ^cd^
Magnesium	29.39 ± 0.95 ^bc^	28.62 ± 0.95 ^a^	26.30 ± 0.95 ^ab^	17.02 ± 2.51 ^c^	17.79 ± 0.95 ^ab^	26.30 ± 2.51 ^bc^
Potassium	350.39 ± 1.64 ^e^	1708.66 ± 0.95 ^b^	2244.70 ± 3.42 ^a^	284.65 ± 3.42 ^f^	729.41 ± 2.51 ^d^	1312.63 ± 4.13 ^c^
Sodium	7.73 ± 0.95 ^bc^	9.28 ± 1.64 ^ab^	11.60 ± 0.95 ^ab^	4.64 ± 1.64 ^c^	12.37 ± 2.51 ^a^	7.73 ± 0.95 ^bc^
Phosphorus	15.47 ± 0.95 ^b^	27.84 ± 1.64 ^ab^	26.30 ± 0.95 ^ab^	12.37 ± 2.51 ^b^	13.92 ± 1.64 ^b^	46.41 ± 28.42 ^a^
Zinc	0.15 ± 0.01 ^c^	0.41 ± 0.01 ^b^	0.39 ± 0.01 ^b^	0.08 ± 0.01 ^d^	0.41 ± 0.02 ^b^	0.78 ± 0.01 ^a^
Manganese	0.18 ± 0.01 ^c^	0.52 ± 0.02 ^a^	0.35 ± 0.01 ^b^	0.05 ± 0.01 ^d^	0.20 ± 0.01 ^c^	0.10 ± 0.09 ^cd^
Copper	0.10 ± 0.01 ^c^	0.06 ± 0.01 ^d^	0.15 ± 0.01 ^b^	0.04 ± 6.01 ^d^	0.11 ± 0.01 ^c^	0.19 ± 0.01 ^a^
Iron	0.06 ± 0.01 ^a^	0.07 ± 0.00 ^a^	0.00 ± 0.00 ^b^	0.00 ± 0.00 ^b^	0.00 ± 0.00 ^b^	0.00 ± 0.00 ^b^
Nitrogen	273.88 ± 0.40 ^e^	397.15 ± 0.54 ^a^	344.79 ± 0.66 ^c^	196.04 ± 0.79 ^f^	357.42 ± 0.40 ^b^	291.95 ± 0.40 ^d^
Na^+^/K^+^	0.005 ± 0.001 ^a^	0.001 ± 0.002 ^b^	0.001 ± 0.003 ^b^	0.004 ± 0.001 ^a^	0.004 ± 0.00 ^a^	0.001 ± 0.00 ^b^

Values indicated are mean ± standard deviation, while letter variations along a column depict significant differences at *p* < 0.05 among samples of *M. sinensis* and *M. paradisiaca* fruits.

**Table 3 plants-08-00598-t003:** Compartmental antinutritional composition of *M. sinensis* and *M. paradisiaca*.

	Samples	Alkaloid (%)	Oxalate (%)	Phytate (%)	Saponin (%)
Banana	Flesh	0.46 ± 0.03 ^c^	17.9 ± 0.07 ^b^	2.42 ± 0.01 ^b^	4.02 ± 0.41 ^abc^
	Peel	0.66 ± 0.34 ^bc^	37.0 ± 4.63 ^a^	2.78 ± 0.33 ^b^	6.57 ± 3.12 ^ab^
	Boiled peel extract	1.76 ± 1.92 ^a^	40.2 ± 5.48 ^b^	2.11 ± 0.02 ^b^	8.12 ± 2.46 ^a^
Plantain	Flesh	0.93 ± 0.00 ^b^	18.8 ± 0.81 ^b^	2.36 ± 0.02 ^a^	1.39 ± 0.98 ^c^
	Peel	0.62 ± 0.11 ^bc^	22.2 ± 0.53 ^b^	2.26 ± 0.04 ^b^	1.16 ± 0.82 ^c^
	Boiled peel extract	0.45 ± 0.15 ^c^	20.0 ± 0.74 ^b^	2.34 ± 0.06 ^b^	2.86 ± 1.23 ^bc^

Values indicated are mean ± standard deviation, while letter variations along a column depict significant differences at *p* < 0.05 among samples of *M. sinensis* and *M. paradisiaca* fruits.

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
