# Peer review of "Comparative Evaluation of the Nutritive, Mineral, and Antinutritive Composition of Musa sinensis L. (Banana) and Musa paradisiaca L. (Plantain) Fruit Compartments"

_plants, 2019, doi:10.3390/plants8120598_

Round 1

Reviewer 1 Report

In this manuscript entitled “Comparative evaluation of the nutritive, mineral and antinutritive composition of Musa sinensis L. (Banana) and Musa paradisiaca L. (Plantain) fruit compartments,” the authors evaluated the several compositions of Banana and Plantain fruit. This study might be conducted in an appropriate manner. My brief comments are as follows:   Comments: 1. In the abstract, the term “AOAC” is not suitable. Please describe its abbreviation. 2. The significance of the treatment condition (40°C for 72 h) is unclear. Please describe this query clearly.

Author Response

Dear sir/ma,

Reviewer 2 Report

General comments

This paper reports results on evaluating the nutritional and mineral composition of the flesh, peel and peel extract components of Musa sinensis and Musa paradisiaca fruits as well as their nutritional and therapeutic potentials. These results are useful for the scientific community. It surely fits into Journal’s aims and scope, and I think it is interesting enough to be published after a minor revision.

Specific comments:

Abstract:

- Please explain the abbreviation – AOAC

- Inductively Coupled Plasma Optical Emission Spectroscopy - after the whole method name you can suggest a shortcut

- World Health Organization - add the shortcut name in parentheses and then use the shortcut in the manuscript only

Introduction

- “Primary metabolites from plants like fatty acids, carbohydrates, dietary fibre and amino acids are central in the developmental and physiological processes in plants, while secondary metabolites (e.g. phenolics, carotenoids, sterols) are involved in plant protection and pollinator attraction.” - I think it should be a reference here.

- “Their use ranges from constipation relief to malnutrition remedy in children, including the restoration of intestinal function and regular bowel movement.” - I am not convinced if this sentence is needed here.

Materials and Methods

- Sample Procurement - it seems to me that in addition to the place (supermarket), the country of origin of the tested bananas should be provided. Were they imported or are they local fruit?

- Data analysis - please explain at which “p” statistical differences were measured. Please also explain exactly between which parameters tested significant differences were measured.

Results

- Please correct table 2, because the values seem to be out of place

Discussion

- Please explain what it is Musa ABB, AAB, and AAA varietal groups.

References

- Please correct the reference position 6.

- Please remove the distance between the position of the reference 60 and 61.

Author Response

Dear sir/ma,
